# Nontoxic Levels of Se-Containing Compounds Increase Survival by Blocking Oxidative and Inflammatory Stresses via Signal Pathways Whereas High Levels of Se Induce Apoptosis

**DOI:** 10.3390/molecules28135234

**Published:** 2023-07-05

**Authors:** Jong-Keol An, An-Sik Chung, David G. Churchill

**Affiliations:** 1Molecular Logic Gate Laboratory, Department of Chemistry, Korea Advanced Institute of Science and Technology (KAIST), Daejeon 34141, Republic of Korea; anjongkeol@kaist.ac.kr; 2Department of Biological Sciences, Korea Advanced Institute of Science and Technology (KAIST), Daejeon 34141, Republic of Korea; aschung40@gmail.com; 3Therapeutic Bioengineering Section, KAIST Institute for Health Science and Technology (KIHST), Daejeon 34141, Republic of Korea

**Keywords:** high selenium, non-toxic selenium, apoptosis, cancer, ROS, inflammation, survival, signaling, COVID-19

## Abstract

Selenium is a main group element and an essential trace element in human health. It was discovered in selenocysteine (SeC) by Stadtman in 1974. SeC is an encoded natural amino acid hailed as the 21st naturally occurring amino acid (U) present in several enzymes and which exquisitely participates in redox biology. As it turns out, selenium bears a U-shaped toxicity curve wherein too little of the nutrient present in biology leads to disorders; concentrations that are too great, on the other hand, pose toxicity to biological systems. In light of many excellent previous reviews and the corpus of literature, we wanted to offer this current review, in which we present aspects of the clinical and biological literature and justify why we should further investigate Se-containing species in biological and medicinal contexts, especially small molecule-containing species in biomedical research and clinical medicine. Of central interest is how selenium participates in biological signaling pathways. Several clinical medical cases are recounted; these reports are mainly pertinent to human cancer and changes in pathology and cases in which the patients are often terminal. Selenium was an option chosen in light of earlier chemotherapeutic treatment courses which lost their effectiveness. We describe apoptosis, and also ferroptosis, and senescence clearly in the context of selenium. Other contemporary issues in research also compelled us to form this review: issues with CoV-2 SARS infection which abound in the literature, and we described findings with human patients in this context. Laboratory scientific studies and clinical studies dealing with two main divisions of selenium, organic (e.g., methyl selenol) or inorganic selenium (e.g., sodium selenite), are discussed. The future seems bright with the research and clinical possibilities of selenium as a trace element, whose recent experimental clinical treatments have so far involved dosing simply and inexpensively over a set of days, amounts, and time intervals.

## 1. Introduction

The biology of animals and humans requires the workings of trace elements such as selenium (Se, Z = 34) for proper physiological functioning. To understand the potential in how Se is properly functioning, we can often turn to what is known as biological sulfur chemistry (S, Atomic number = 16), its function, and metabolism in biology. The trace concentrations of selenium that exist in biology and their bio(macro)molecules have precise purposes in redox biology. Selenium can exist as a reactive center at particular places in biomolecules, allowing the enzyme to function as an antioxidant, anti-inflammatory reagent, and in an immune-related capacity (e.g., boosting immune function). Additionally, there are investigations that describe selenium’s role in cancer incidence reduction (see the discussion below); in particular, the reactions of selenium can help block metastasis in a limited number of published clinical cases [1,2]. Despite being a trace element, selenium and the concentrations of Se-containing small molecules are essential; again, these atomically precise systems can relate to both oxidation and reduction processes in living systems.

Our understanding of all aspects of selenium in biology and medicine continues to develop through further research and normative amounts of Se are well known. In addition, we should state that the normal range of selenium in the blood is about 120–160 micrograms per liter (mcg L^−1^; µg L^−1^). Additionally, the maximum intake is ~400 micrograms per day (Table 1). The effects of excess selenium intake are, in some cases, salubrious, as discussed below in certain cancer studies. Cancer is a serious progressive and commonly fatal disease of dire human concern and requires different medical interventions and often surgery.

Selenium in cancer would relate to the action of selenium-killing cells. Therefore researchers (and clinicians) continue to further explore the biological and chemical underpinnings of apoptosis. (Note: We also provide a separate section about ferroptosis and senescence below). Apoptotic events can involve oxidative stress; certain disorders would naturally give rise to a perhaps prolonged state of oxidative stress.

There is, after all, a collection of small molecules that make up the molecular composition of oxidative stress, often observed as a cascade reaction composed of a variety of low-MW species. Examples of well-known reactive oxygen species (ROS) are hydrogen peroxide (H_2_O_2_) and superoxide (O_2_^−^) [5]. 

Selenite is a well-known small molecule containing one central Se center, a small species that is known in biology and which can be administered as a treatment. Physiological concentrations of selenium in biology introduced at a level of 1–3 µM were found to increase certain cell processes which are tracked and monitored. The experiment detected an increase in cell proliferation. The activation of glycolysis, the maintenance of constant mitochondrial membrane potential, and BCl-2 upregulation led to increasing glucose uptake and adenosine triphosphate (ATP) generation. The aforementioned phenomena are regulated by Ak strain-transforming (Akt) activities; according to our current understanding, introduced selenite stimulated PI3K/Akt pathway activities. Some of these ideas are embodied in Figure 1.

In recent research reports, it was found that selenite may act on sulfhydryl residues within a protein complex called apoptosis signal-regulating kinase 1 (ASK1). However, such reactions are reversible; agents such as dithiothreitol and β-mercaptoethanol are sulfhydryl-reducing reagents that can be introduced and cause a chemical reduction in these sulfhydryl-containing groups of ASK1 and other systems. The reagent disulfide groups become reduced, as we see in biological Cys reduction from the disulfide linkage [7]. These reagents above help reduce sulfhydryl and, as a result, convert to their oxidized forms. 

The notions above are also related to chemical stress and the cell’s attempt to survive against this stress. When apoptotic signals are inhibited and anti-apoptotic signals are enabled, cell survival and proliferation will occur and act against external stresses; selenite is thus found to positively impact biological systems.

These phenomena surrounding selenium biochemistry constitute the Se version of cellular defense. Conditions such as neurodegenerative diseases, acute respiratory distress syndrome (ARDS), acquired immune deficiency syndrome (AIDS), and oxidative stress have been reported to be ameliorated. Further exploration of the effects of Se and Se-containing small molecules on biology is ongoing; further findings would be essential to the trace element research community. One future goal in research would be to check the relationship between the induction of Akt activity and the reduction of the ASK1 complex (see future aims below).

## 2. Selenium Compounds and Their Metabolism

In human and animal diets, selenium is taken in by ingestion/digestion in organic and inorganic forms (Figure 2 and Figure 3) [8]. Common organic selenium-containing compounds include amino acids/peptides such as selenocysteine (SeC); SeC is considered the 21st encoded amino acid. Selenomethionine (SeM) can be misincorporated into proteins in place of methionine (Met) or can be formed post-translationally. Methylselenocysteine (MSeC) is another amino acid under discussion in the context of the presence of selenium in peptides [9,10]. Inorganic forms are also commonly considered in health, nutrition, biology, and biochemistry. The commonly encountered inorganic forms include selenate and selenite. From a food nutrition standpoint, MSeC is abundant in certain food sources. Brazil nuts, broccoli, garlic, walnuts, and other plants are rich in MSeC [11,12]. Some molecules are more active/reactive than others. The most chemically reactive species containing Se in a biological context is hydrogen selenide (H_2_Se, produced from selenide biologically), methylseleninic acid (MSeA), and methylselenol (MSeH) (note: biological MSeH is produced from MSeC). These trace metabolites are important because they produce ROS to the extent that they bear antitumor capacity. Thus, these Se species can indirectly fight cancer, wherein the intermediaries in this biological battle are the ROS (introduced above). The ROS produced can chemically oxidize sites in biomolecules, especially sulfur centers in sulfhydryl groups. The reduced form of glutathione (GSH) possessing a cysteine (Cys) residue is easily affected by ROS levels. The current view of ROS from a free radical chemistry perspective is that ROS are often good and essential in processes such as signaling and fertilization. ASK1, for example, may also be oxidized in this way (see below). Selenodiglutathione is also an important species that should be introduced. The reaction of GSH with selenide will produce H_2_Se by way of selenodiglutathione. H_2_Se induces nuclear damage: DNA methylation (Adenosine-Methionine). Other effects of H_2_Se include ROS production and cytotoxicity (cell apoptosis). However, H_2_Se and MSeH can also be viewed as constituting building blocks in biology. These can serve as the central foundation for glutathione peroxidase (GPX) and thioredoxin reductase (TrxR) because these small precursors will be used in forming selenocysteine (SeC). A description of selenium metabolism is shown in Figure 3 [8]. We should also mention the Se and redox balance: the redox balance in biology (cells) will depend on [H_2_Se] and [MSeH]. One atom of selenium can also undergo a series of methylations. That is to say, R–Se–R can become methylated to produce R–Se(Me)(R)^+^ [13]. These covalent modifications to the central selenium center impose a large change to the molecule yet are found to be reversible in biology. 

As with other amino acids (AAs) in proteins (peptides), SeM can covalently become part of the protein and active side, incorporating and bearing the Se-containing residue (sidechain). Because of this incorporation, SeM is less able to be active or metabolized to selenite or other products such as MSeC. In our studies, when we monitored the occurrence of apoptosis, tumor invasion, and metastases, we found that MSeC is most effective when measuring the induction of apoptosis; moreover, tumor invasion and metastasis were observed to be blocked [14,15,16]. To better understand elimination from the body, it will be necessary to carry out further in vivo studies of MSeC to assess the maximum tolerable dose of MSeC and the half-life of the concentrations of MSeC in human biology (pharmacokinetics). Furthermore, MSeC is commercially available as over-the-counter supplements (Life Extension, etc.). Therefore, the application of effectively treating cancer in humans with this chemical would be an interesting investigation.

In Figure 4, we depict ROS formation from the presence of selenium. H_2_S, produced from MeSeH and selenide, can react with sulfhydryl groups, including those of GSH; the result is that O_2_^−^ and H_2_O_2_ are produced. Apoptosis occurs from the blockage of the S/G2 Phase caused by the arrest of the cell cycle; this occurrence was created by DNA single-strand breakage caused by this Se-induced ROS that can chemically oxidize DNA at the atomic level [17,18,19]. The condition of leukemia, for example, and its treatment can be studied with selenide induction of apoptosis of these cells with the effect of ROS and selenium on DNA in mind; proapoptotic and genotoxic effects were found to be present by the production of O_2_^−^ in these clinical leukemia medical cases. Regarding the proposed mechanism, superoxide dismutase and related enzymatic mimics [20] can block cancer cells (mammary and prostate); when a hydroxyl radical scavenger is used, the cancer is not blocked [21]. Cell death by selenite action was found to be blocked when catalase was used and treated in the cell culture medium [22]. In certain cell lines, cell death was observed—both cell lysis and apoptosis—when selenite was added. Selenite is able to be metabolized to H_2_Se; H_2_Se will target DNA and break DNA single-strands [21,23,24,25]. “Death proteases” were not found to be related to cell death induced by H_2_Se [14,26,27,28]. The action of MSeH and its providing selenium is anticarcinogenic, as has been described and discussed in the literature [29,30,31,32,33,34,35]. 

Apoptosis induction and G1 Phase and mammary cell growth inhibition have been demonstrated with MSeC and methylselenocyanate (MSeCN), the precursors of MSeH [18,19,27,36,37,38,39]. Apoptosis is caspase-independent [28]. Caspase activity and the membrane of mitochondrial potential were induced and altered by ROS production, followed by apoptosis, which, in this study, was induced by MSeH. ROS leads to Cytochrome c release; this action helps active caspases. Here, caspase 3, caspase 9, and caspase 8 were induced, thus helping achieve apoptosis.

Further, the cellular enzymes contain catalytic clefts; in these domains, the sulfhydryl groups were oxidized, directly induced by the presence of MSeH [40]. A recent review paper helps carefully describe the cytotoxic effects of H_2_Se, and MSeH; the MSeH species gives rise to H_2_Se, and MSeC gives rise to MSeH [41]. The methylation of Se can formally include a stepwise process (vide infra). MSeH can lead to the formation of elemental selenium. MSeH can form dimethyl selenium and other small-molecule selenium-containing compounds.

The action of methyltransferases and demethyltransferases can allow for the addition or release of methyl groups [13]. Selenium is excreted in several preferential forms, some of which are *N*-containing; these small molecules include seleno-metho-*N*-acetyl galactosamine, trimethyl selenium, and seleno-methyl-*N*-acetyl glucosamine [42,43].

## 3. Signaling Pathways and Molecular Targets Pertinent to Biological Selenium

Regarding anti-cancer effects, there are signals, metabolic pathways, and molecular targets that we seek to better understand for the purposes of developing better cancer therapies; despite what we think we now currently know about MSeH, scant information is available about these processes, so more detailed studies are required to accrue more precise results.

Metastatic colonization involves an essential extravasation process in which endothelial and circulating tumor cells interact. Focused cell adhesion involves Talin 1, which deposits fibronectin (endothelial) to which cancer cells are attached. Reduced liver metastasis is seen to result in studies when transendothelial migration and cell adhesion are impaired when Talin 1 is depleted [44]. B16F10 melanoma cells were studied and found to reveal anti-cancer function when MSeH is made available in sublethal levels; MSeH is generated from SeM and SeM-METase (methioninase). Apoptosis arising from caspase action is observed due to the alteration of adhesion characteristics stemming from integrin expression modulation; this is, therefore, the proposed mechanism of apoptosis [44,45,46]. Integrins are well-studied receptors: Integrins mediate cell adhesion, are a significant family of receptors at the cell surface, and are linked to specific extracellular matrix (EXM) proteins. The family of integrins can also be introduced into our discussion: integrin ligands fibronectin (FN) and vitronectin (VN) can be tracked through this research. Integrin expression relates to Talin 1 levels, and [Se] introduction may reduce Talin 1 levels, but this issue, as with others, should command more study by research groups. Metastasis, invasion ability, and tumor growth are believed to crucially derive from integrins and extracellular matrix (ECM) interactions [47,48,49,50,51,52,53].

Matrix metalloproteinases have also been a focus of ion selenium-related biological studies. HT1080 cells are inhibited by selenite. MMP-2 (Matrix metalloproteinase-2) and MMP-9 (Matrix metalloproteinase-9) are suppressed by HT1080 cells when acted upon by selenite [25]. 12-*O*-Tetradecanoylphorbol-13-acetate (PMA) induces pro-MMP-2 activation, and tumor invasion was blocked by MSeA treatment. This finding is comparable to the earlier study in which MSeH was used [54]. A study from 2001 showed that pro-MMP-2 activation is correlated to the production of type-1-MMP (MT1-MMP) (membrane type) [46]. Our work demonstrates that pro-MMP-2 activation requires MT1-MMP as an essential factor and that MSeA blocked MT1-MMP in a dose-dependent fashion. Integrin expression is the mode whereby NF-κB regulates MT1-MMP expression. The reduced expression of integrin involved NF-κB activity suppression, which allowed for MT1-MMP to be affected by MSeA. Upon introduction of [MSeA], ROS production was found to be decreased; PMA induces ROS production. The inhibition of pro-MMP-2 activation comes from the NF-κB signal and arises from the suppression of MT1-MMP expression; the presence of MSeA blocks tumor invasion (Figure 5).

Additional species such as METase, SARS-CoV-2, AREs, and NRF2 have been discussed in the biological selenium literature. Methylselenol can be produced by SeM-METase, which also confirms the notions of these experiments [45]. Pasini et al. recently (2021) reviewed antioxidants and therapies that are anti-inflammatory in the context of SARS-CoV-2 complication prevention [55]. Regarding detoxification and antioxidant enzymes, one transcriptional factor that pertains to gene coding is the p45-related factor 2 (NRF2) which is a nuclear factor erythrocyte 2 [56]. Heme-oxygenase is one example of a gene classified as an antioxidant response element (ARE) which can be regulated via induced and basal expression by NRF2. Enoyl-CoA hydratase-associated proteins are Kelch-like (Keap-1), serving as a negative regulator that blocks NRF2-dependent transcription in basal conditions. Target gene expression is driven forward when a nuclear accumulation of NRF2 exists; this happens when electrophiles and oxidative stress are present in cells [57]. Injuries to cells may arise from oxidative stress—an imbalance between oxidative and antioxidative events. Evidence that viral infections can participate in this is supported by various data in published studies [58,59,60,61]. Oxidative injury and inflammation result from the activated NF-κB transcription factor and inhibition of the NRF2 pathway caused by respiratory viral infections; upregulation of ARE gene expression and NRF2 activation are induced by pro-oxidations [62].

The release of viruses and cell death involve propagation mechanisms that can be particular and depend on NRF2 activation/inhibition. The stage of the infection also depends on NRF2 activation/inhibition [63,64]. Patients suffering from SARS-CoV-2 infections received biopsies and statuses in which the NRF2 pathway was repressed; this evidence demonstrated that the essential cytoprotective pathway is deprived in SARS-CoV-2. Inflammation regulates biochemical pathways in ROS-mediated NF-κB; this can be prevented by adding antioxidants such as *N*-acetylcysteine, which attenuates oxidative stress and reduces the action of NRF2 [56,57]. NF-κB translocation is impeded, and NAC (N-Acetylcysteine) activates p38 phosphorylation, which results in the restoration of total thiol intracellular concentration and intracellular [H_2_O_2_] decrease [65].

When inflamed tissues feature innate immune cells, NRF2 and NF-κB undergo some reciprocal crosstalk [66,67,68]. Human macrophages and mice lungs feature an NF-κB switch on a vent in SARS-CoV-2 infections, according to in vitro studies [69,70]. Contrastingly, in SARS-CoV-2 mice models, survival is ameliorated, and inflammation is decreased upon NF-κB inhibition. With patients with SARS-COVID active infection, the NF-κB may be downregulated by specific activation of drugs. Despite NRF2 suppression being linked to NF-κB and ultimately with inflammation, Keap1 (Kelch-like ECH-associated protein 1) can be targeted by pharmacological inducers that can trigger NRF2 repressor function, which can be disabled on Keap1. Electrophiles such as dimethyl fumarate (DMF) and sulforaphane can induce an effect as well; they are referred to as NRF2 inducers, and thus, they act to disable Keap1 [71]. An inactive form of NRF2 exists when NRF2 is bound to Keap1. Virus replication and transmembrane series 2 protease (TMPRSS2) are reduced when Keap1 is deactivated and NRF2 is induced.

In Figure 6, in the case of SARS-CoV-2 infections, NRF activators are shown [55]. SARS-CoV-2 complications of the severe type can present when TMPSS2 activity, NF-κB activity, and viral replication are reduced when the inactive form of NRF2 is present (when NRF2 and Keap1 bind together). As extensively discussed in the Pasini et al. 2021 review, the suppression of NRF2, which happens upon SARS-CoV-2 infection, is related to the aforementioned TMPRSS2 activity, NF-κB activity, and increased viral replication. For conjecture about Keap1, please see the Conclusions section.

In our research and that of others in the future, the NRF2 activators and Keap1 modification incurred by NF-κB activation would be a great focus of study [55]. As far as selenium involvement is concerned, it would be important to investigate whether selenium-containing compounds may be involved in altering NRF2 activity; this could be achieved by reducing NF-κB or modifying Keap1 (see the Future Aims section, vide infra).

In our previous work, the induction of inflammation and decreased integrin expression was observed in the treatment of PMA in cancer cells; NF-κB activity was reduced by selenium administration. For SARS-CoV-2 infection, which is more severe, the potential role of antioxidant and anti-inflammatory therapies can be considered [55]. Therefore, the NF-κB signals we determined by their point in a similar direction as such toward antioxidant therapy.

## 4. Anti-Inflammatory Activity and Immune Functions of Selenium Addition

Immune cells have many commonalities with other types of cells. Immune cells respond to nutrient levels, such as the concentrations of selenium provided by the organism’s diet. As a result, selenoproteins are expressed. Interestingly, some selenoproteins are more abundant than others. For example, GPX1 and GPX4 were found to be increased upon supplementation of sodium selenite (NaSeO_3_); the supplemental level was 50 or 100 µg per day [72]. This earlier laboratory research also serves as a good basis for more recent related proteomic research, published, e.g., in 2018 [73]. IL-2 (Interleukin-2) receptor activation enables immune function upon the addition of Se-containing compounds [74]. Immune cells are differentiated by the immune system, which is activated by IL-2, which is activated by T helper cells. Immune cells are protected by the scavenging for ROS that can be enabled by selenoproteins whose Se centers are antioxidant functioning in nature. Immune functions can be regulated by immune cells activated by the IL-2 receptor that the Se activates. Anti-inflammatory and antioxidant functions are exhibited by selenium. The roles of selenoprotein P (Seleno P) and W, TrxR, and GPX impart such clear functions of their enzyme that we now understand their function and mechanisms more completely. There may be more functions of these enzymes that can be elucidated and published in the future. The tumorigenesis, apoptosis and differentiation, and redox regulation of cytokine and antioxidation defense can be grouped together for our discussion: These physiological processes can be altered by selenoproteins as treated and focused in the previous literature reviews [75]. Phospholipid peroxidase and H_2_O_2_ can be reduced by glutathione-peroxidase. The production of leukotrienes and inflammatory prostaglandin production is much reduced by cyclooxygenase and lipoxygenase pathways, and H_2_O_2_ intermediates, as well as ROS and free radicals, which are reduced by the presence of phospholipid peroxidase and H_2_O_2_ [76]. In the case of asthma, pancreatitis, and rheumatoid arthritis, for example, these conditions are affected by inflammation or oxidative stress. Therefore, they are possibly controllable by the administration of certain selenium-containing compounds. Patient joints were affected. Pain ensued, and the pain was reduced (a study of patients with rheumatoid arthritis). In this study, selenium was administered in the amount of 200 µg [77]. Frequent attacks of discomfort and pain were reduced in studies of patients with pancreatitis when Se was administered. In this study, the Se was administered in doses of 600 µg, and the pancreatitis cases who were treated had both the types of recurrent and chronic [78]. In an occupational case study with a control group, selenium intake was shown to correlate with asthma; these important diet supplements can provide a protective effect [79]. Phagocytes can kill microbes which are associated with the production of ROS by immune cells. For complete and effective immunity, the defeat of the phagocytosed pathogens, macrophages, and neutrophils must generate ROS. Various non-phagocytic immune cells and phagocytes communicated by (i) cell–cell interactions and (ii) cell signaling events are now better understood as being conducted by way of ROS concentration-related phagocytes that produce a burst of oxidation, yet are disrupted by the function of enzymes when they have genetic differences that lead to oxidative buildup. Fungal and bacterial events may occur in certain severe cases and even possibly life-threatening cases when these arise from granulomatous disease (chronic granuloma) [80]. When phagocytes cannot completely eliminate pathogens, there may be recurrent infections; this is true with diseases such as those that present chronic inflammation. However, independent of the incidence of infection, there may be present persistent inflammation. Terminating immune responses in, e.g., an autoimmune disease requires that NADPH oxidase (NOX2) should be working properly [81]. The deficiency of NOX2 will create autoimmune disease and hyper-inflammation responses. Their signaling and functions involve secondary messengers such as ROS in various immune cell types, including immune cells. The NOX2 case is a clear case of how immune cells can become activated and how selenium compound supplementation is a logical strategy. Other processes may also clearly show, in the future, how messenger ROS are essential. Without question, supplemental Se provided from donating can affect the ROS level; ROS production naturally has many downstream effects. It is common to consider an oxidative cascade. Nonphagocytic and phagocyte cells will provide an oxidative burst; this is directed by these levels present in immune cells. Zymosan particle opsonization or PMA are stimulants pertinent to and utilized in these studies. When these are incubated with neutrophils (rats) for prolonged time periods, there is a reduced incidence of oxidative bursts. The rat-model animals involved in these studies were, in fact, Se-deficient [82]. NADPH-dependent superoxide-generation, when lowered, is linked to inadequate metabolism of ROS, such as H_2_O_2_, therefore causing an oxidative burst. Studies of the neutrophil can support the idea that Se-containing proteins are used in this regulation. Therefore, the strength of the oxidative burst and the H_2_O_2_ concentration are symptoms of an important feedback mechanism. IgG (Immunoglobulin G)-protein is seen to be opsonized when phagocytosed by macrophages; the macrophages exhibiting a decreased burst of oxidation were found to be true for studies of Selk-/mice models [83]. The oxidative burst process can be affected by the effect of Se supplementation on cells (immune); the level of Se (1.0 ppm Se) can also affect the oxidative burst process. This is made possible through the action of peroxiredoxins and GPX. As a secondary messenger, H_2_O_2_ can provide two alternative mechanistic models. Two alternative models of action regarding peroxiredoxins and GPX serves as a messenger: (i) The conventional model involves the activation state and active site conformation being altered because of the S_cys_–S_cys_ oxidation, formation of a cysteine group and a Cys crosslink in which the two cysteines are within the same protein; (ii) the updated model involves adjacent disulfide bonds, which can be individually acted upon by peroxidase enzymes, such as the GPX1 enzyme, and which may act upon the disulfide bond.

By a delay of providing oxidants to signaling molecules, the action of H_2_O_2_ oxidative capacity can be promoted, and therefore the peroxidases do not minimize oxidative signaling by H_2_O_2_ action [84,85]. One yeast example involves the GPX–Yap disulfide bridge created from the action of the GPX3 transferring oxidative equivalent [86]. Proliferative capacity can be increased by a cluster of differentiation 4-positive T lymphocyte (CD4^+^T) cells by Se supplementation; a supplement can increase TCR (T cell receptor) [87], translocation, activation of NFAT and transcription and oxidative factors, and Ca^2+^ flux can be enhanced, all of which are parts of the TCR signal. These TCR signals can induce IFN (Interferon)-gamma gene expression and IL-2 activity. The high free thiol content involved in these signaling events in the mechanisms is tied to Se levels. Dietary Se levels, when increased, will increase the levels of free thiols in vivo [87,88] when N-acetylcysteine is added as a source of free thiol into cells, eliminating the proliferative capacity and TCR-induced Ca^2+^ flux [87]. However, yet to be identified are specific changes in specific disulfides during this process. When observing the stimulation of immune cells, the addition of light intake levels of selenium does not attenuate the level of ROS in these cells or their expression of selenoproteins [89]. The first oxidation is receptor-mediated; the half-life of this reaction can be measured and compared, and regulation of this reaction is demanding on selenoproteins such as GPX1. The signal strength of T cells is contributed to by selenoproteins. Redox signaling in immune cells and calcium and Se level dependencies are described below and appeared in the literature [89,90,91,92]. The signaling molecules or redox intermediates are directly acted upon by the oxidative “burst,” which is a receptor-mediated action and may involve downstream effects emergent from the involvement of selenoproteins [74,87,89]. Anti-apoptotic signals and apoptotic signals are inhibited when selenite is added; in this way, the selenium compounds ultimately contribute to proliferation against excessive oxidative stress and contribute to cell survival. Neurodegenerative diseases, ARDS, AIDS, and oxidative stress can be studied in their (indirect) being combated with Se compounds upon administration. This is why Se is beneficial to cells and helpful toward a good “defense” for cells. In our studies, the induction of Akt and the reduction of ASK1 were found to be related and supported so that the cell survival signals relate to the presence or addition of selenium.

## 5. The Role of Selenium in Senescence and Ferroptosis

In this section, we describe recent developments about selenium involvement in ferroptosis and senescence. While the previous sections have provided an excellent foundation, the idea of the selenium effect focusing solely, or mainly, on apoptosis is arguably outdated, and the science of selenium-induced apoptosis in the literature remains fragmented. To control cell growth/eliminated unfavorable cells (e.g., cancer/precancerous cells), we introduce results in the last decade and a half below that are illuminating; research has unearthed at least two new pathways: senescence and ferroptosis. For senescence, examples of research reports include those from Cheng and coworkers [93] and [94]. These findings help the Ataxia telangiectasia mutated (ATM) pathway, senescence/nano-selenium species, and redox regulation to be placed into context.

In a successful senescence study by Cheng and coworkers from 2010 [93], a study over a course of 1–7 days two different types of Se-containing compounds were tested in cancer cells and in normal control cells. Both inorganic and organic forms of selenium were tested in this report: sodium selenite, MSeA, and MSeC were introduced. The cells were treated over a 48 h period, but then there was up to a seven-day recovery period. The types of cells used were as follows: normal lung fibroblasts (MRC-5), normal colon fibroblasts (CRL-1790), prostate cancer cells (PC-3), and colon cancer cells (HCT-116). Several senescence-related reagents and methods were employed to seek a better understanding of how selenium behaves in these contexts. These include the disease pathway that seeks DNA damage by ATM action, levels of the 5-bromo-2-deoxyuridine substrate (BrdUrd), and beta galactosidase, which is senescence-related.

The next article was a study authored by Cheng and coworkers from 2010 that deals with colorectal cells [94]. As with the above study, sodium selenite, MSeA, and MSeC were introduced. ROS was produced as a result. A mismatch “repair protein” was identified as being operational in this study. The selenium-containing compounds, when introduced, help activate a repair response. The DNA damage repair is ATM-dependent. Since the MMR route is targeted, the findings signify a new role for selenium compounds and for biological/medicinal hopes for selenium.

The same expansion of what we used to understand as apoptosis is true for ferroptosis. These notions are clearly expressed in reports by Conrad and Stockwell and their coworkers [95,96,97]. A report by Stockwell and coworkers in 2022 (an article with 15 different professional affiliations) allows us to consider ferroptosis from a variety of data collected and in which the authors try to advance the quest of “precision therapy” [95]. The mutations of GPX4 include an R15H variant that is found in three different individuals who suffer from a condition known as Sedaghatian-type spondylometaphyseal dysplasia (SSMD). Thus, this study is, on the one hand, a meaningful thrust in understanding ferroptosis, and is also, on the other hand, investigated in the context of a rare genetic disease.

The next article, a short contribution by Ingold and Conrad in 2018, tries to answer the age-old question about the importance of the choice of the element selenium in redox biology by exploring, e.g., the sulfur version and testing this variant [96] (Figure 7). The authors propose that if cysteine is present instead of selenocysteine, then what happens to the mammalian systems can exemplify the need for Se. It was found that GPX4 prevents damage to the fatty acid oxidation of the phospholipids and therefore protects the nervous system of developing animals. The sulfur version of the enzyme, however, becomes overoxidized and becomes rapidly oxidized. This process is not reversible. Because of this elemental center oxidization which is iron and ferroptosis dependent, as studied for GPX4, selenium can be tied to ferroptosis!

The third and last article that we can describe in this short section about ferroptosis is also by Marcus Conrad and coworkers (from 16 different professional affiliations), which focuses more on the hydrogen peroxide angle of the research but relates to the previous report in that GPX4 is produced [97]. Figure 7 below shows the involvement of lipid peroxidation, which results from an inactive enzyme; specifically, the comparison that was made in this case is one where the replaced active site, where sulfur is present and will oxidize irreversibly to the sulfinic or sulfonic acid.

There are some common biomolecules affected by oxidation and reduction, which indicate that DNA base pairs might also be able to undergo chemical reduction. Repairing this oxidation is of great importance in biology. Base pair variants such as 8-oxo-7,8-dihydroxyguanosine and 8-oxo-2′-deoxyguanosine (8-oxo-dG), for example, are two specific biomarkers that illustrate ROS have been active. Moreover, there might be 20 or more different oxidation products alone relating to DNA constituents/fragments [98]. The role of flavoprotein thioredoxin reductase as an antioxidant is of great interest [99]. If so, could it serve as a reductant for oxidized DNA bases? For example, a paper about thioredoxin (Trx) by Kamal et al. discussed leukemia [100]. In this paper, Trx levels were studied. Additionally, 8-OHdG (8-Hydroxy-2′-deoxyguanosine) could be monitored. The section titled “Expression levels of Trx was negatively correlated with serum 8-OHdG and TM (Transaction Monitoring) in acute myeloid leukemia (AML) and acute lymphoblastic leukemia (ALL) groups” shows the statistics about this important finding to indicate that thioredoxin is able to reduce DNA. For thioredoxin reductase, however, being able to reduce oxidized DNA, such a detailed report of how TrxR can be used to reduce all of the oxidation products of DNA base pairs has yet to be published [101]. However, thioredoxin reductases have been discussed in the context of DNA repair, but as for extensive repair of oxidized DNA fragments, this is both a future goal of great importance and difficulty.

## 6. Benefits of COVID-19 Se Treatment and Related Sepsis Conditions

It has been found that cancer, diabetes, and hypertension, as well as chronic obstructive pulmonary disease (COPD), are comorbidities of coronavirus disease 2019 (COVID-19) sufferers; the elderly are especially greatly affected by the novel severe acute respiratory syndrome coronavirus infection (COVID-19) during its recent prevalence [102,103]. Massive chemokine and cytokine release are exhibited by the immune system when overreacting during severe cases of the disease [104]. COVID-19 infections have threatened the world’s public health and are highly transmissible. ARDS (acute respiratory distress syndrome), hypoxemic respiratory failure, and septic shock can occur in severe cases of COVID-19. Severe versions of COVID-19 usually present as pneumonia and fever, among other symptoms. Intensive care may be complex or prolonged as profuse inflammation and distinct immune-related changes are seen to occur in addition to the conditions discussed above. Lymphocytopenia, or immunological changes, were also studied in the context of selenium addition [105,106]. Se status, when measured and tracked, has been shown to be inverse to sepsis susceptibility, which serves as a risk factor for mortality in COVID-19 [105]. An individual’s organ damage can result when a vicious cycle occurs between the interplay between oxidative stress and inflammation [107]. ARDS, a severe disease symptom, commonly involves antioxidative defense systems that rely on essential micronutrients such as zinc and selenium. Parameters that are elevated for inflammatory values, as well as depressed levels of zinc and Se (biomarkers), were found for most patients upon their being tested during hospital admission. GPX3 activity can be further increased. Selenoprotein P and Se levels can be significantly increased to normalized levels upon selenium supplementation. IL-10, IL-1 beta, IL-6, PCT, and CPR are inversely correlated with Se biomarkers, particularly SELENOP levels (Figure 8). Total IgG, NK cells, and CD8^+^ were positively associated with Se biomarkers.

Regarding the outcomes, non-survivors tended to have lower Se levels than survivors. Therefore, for severe cases of ARDS stemming from COVID-19, for an adequate immune response, sufficient Zn and Se levels appear to be essential. ARDS patients were studied. Two groups were studied, one which was a control saline-treated group and one that was a Se-treated group. Il-6 and IL-1 beta were inversely correlated with increased Se levels upon treatment with NaSeO_3_. Further, the treatment correlates with reduced FRAP (ferric-reducing antioxidant power) and increased GPX activity [108]. Respiratory mechanics were improved, inflammatory stress responses were moderated, and this selenium intervention appeared to help restore lung antioxidant capacity. Despite these changes, during the hospital stays (intensive care unit stay), the duration of ventilation by mechanical means and the overall survival rate were not altered by selenium intervention. The possible positive clinical effectiveness can be observed upon completing, summarizing, and analyzing more clinical trials when they become available [105]. In another report, the COVID-19 mortality risk was found to be associated with Se deficiency. The need to support recovery, strengthen the immune system, suppress virulence development, and reduce infection risk can be addressed by requiring safe and promising, as well as fast, means to combat COVID-19 because of the pandemic’s recent universal threat to humanity. We propose that Se is the most relevant and interesting trace element for tracking these issues. Dietary additions or supplements can be provided for patients dealing with COVID-19 infection, as well as preexisting conditions (so-called comorbidities) or those sufferers enduring a more prolonged course of the disease and living in a Se-deficient environment or eating sub-nutritional food. High research standards will have to be involved in intervention studies that better scrutinize a proposed underlying feed-forward mechanism because of our observed association of Se deficiency with mortality risk. Safety and limited expenses are suitable for these selenium-related adjuvant treatments.

Se supplementation with patients enduring ARDS can be sought, as well as other clinical applications which might be accessible and therefore applicable. A hypothetical mechanism is proposed for Se supplements that could then prevent COVID-19 [109]. The virus cannot penetrate healthy cell membranes if its protein disulfide isomerase is oxidized by sodium selenide; moreover, sodium selenite can oxidize thiol groups in peptides such as GSH. From an inorganic chemistry perspective, selenite with its tetravalent cation (Se^4+^) can accept two electrons to become a divalent cation (Se^2+^) and, in so doing, can act as an oxidant. In terms of antiviral properties, this is one implication of selenium and its oxidizing capacity. We already discussed the practical differences between selenium and sulfur above. According to the following chemical equation, the viral protein isomerase (PDI) can be represented to be converted regarding their sulfhydryl groups at the active site:PDI − (SH)_2_ + Se^4+^ → PDI − SS + Se^2+^

Based on the product formation, when the reaction goes to completion, there are no sulfhydryl groups left to chemically react with [110]. Healthy cytoplasm entry is prevented when the virus does not possess free sulfhydryl groups that would ordinarily be available for reactions. The so-called hydrophobic domain is incapable of undergoing the necessary reactions. The contribution of ARDS is significant to morbidity and mortality, which is discovered in critically ill patients and is observed as a frequent complication [111]. Expired gases and bronchoalveolar lavage are used to collect specimens to help detect data on oxidative lung injury, a key indicator of ARDS; ARDS involves the action of metalloproteinase enzymes and myeloperoxidase [112,113]. In ARDS cases, the patient’s plasma bear markers of lipid peroxidation resulting from oxidative stress; reactive oxygen species and reactive nitrogen species were also detected as a secondary wave [114,115]. ROS is engaged by GPX protein family members; thus, pulmonary antioxidants help protect against oxidative injury as a first line of defense [116]. IL-6 and IL-1 levels decreased, but GPX3 was increased in serum when levels were tested upon infection and intervention with NaSeO_3_, after which treatment Se levels increased [108]. Septic shock and sepsis (severe) cases were observed in patients; these cases were administrated with a large dose of NaSeO_3_ as an adjuvant treatment that reduced mortality [117] (Note: GPX and TrxR as selenium enzymes should be separated/distinguished from glutathione and thioredoxin (as small redox proteins) in the regulation of selenium metabolism in the cell. For a more detailed treatment, kindly see the reference by Burk and colleagues [8]). Notably, in the study by Angstwurm et al. (2007), despite administering high doses of sodium selenite, there were no observed side effects. When Se deficiency is manifested, supplemental treatment with Se may be sought. In the case of COVID-19 has, up until recently, posed a grave and universal threat to human health. Thus, the administration of Se may help in (i) supporting recovery, (ii) bolstering the immune system, (iii) reducing the innate virulence development of the virus, and (iv) reducing infection risk in a safe, rapid, convenient, and inexpensive means. However, further research will be required.

For a properly functioning immune response and overall human health, Se is not only important, but it is also an essential (trace) element (vide supra). Se “status” (selenium can be measured from drawn blood, and normal levels are stated to be 110 to 165 mcg L^−1^); these values are inversely related to mortality risks from severe diseases such as polytrauma or sepsis. An immune response may be diverted in inexpensive manners, such as by adding Se, and having this careful administration serves as an essential guide for clinical studies with Se, according to certain published studies on human subjects and also on mouse models. The respiratory tract, including the lungs as the target organs and the immune cells, must be subjected to further clinical studies before Se supplementation can be concluded as being beneficial. Clinical treatments of advanced cancer patients in which a high amount of Se treatment was used, as well as anticancer function, immunity, aging-related diseases, and aging, were carried out. There is a need to carefully analyze the results from these studies for further planning. In all of these cases, an essential factor is the existence of selenium status [118,119]. Anticancer functions and further high amounts of Se treatment with advanced cancer patients have also been presented and studied [2].

## 7. Clinical Applications of High Doses of Sodium Selenite (Cancer Studies)

Cancers can be targeted efficiently by the use of selenium alone. Radiation therapy and adequate chemotherapy have interestingly been used to give the otherwise ordinary addition of selenium a more pronounced effect [2]. Before radiation therapy, administration of NaSO_3_ in a one (single) dose orally (2 h before therapy) to metastatic cancer patients (5.5–49.5 mg of Se compound). It was shown that sodium selenite was effective in increasing the ordinary effectiveness of radiation therapy and ameliorating the downsides of radiation treatment: side effects such as pain were reduced. The cancer status/staging was stabilized, and further, PSA in prostate cancer patients was reduced [120]. When up to 33 mg per dose of selenite was administered, no adverse effects were shown. Up to a 33 mg dose was reached, and treatment of patients in a palliative fashion and with metastatic cancer with NaSO_3_ was found to be safe and efficient, as undertaken in phase 1 clinical studies. A similar reported study involved the treatment of selenium with four weeks or two weeks of administration in which the subjects were given selenite for five consecutive days, and these patients had chemo-resistant tumors [121]. In the patients in these studies, the legs and fingers were reported to be locations that experienced cramping; nausea and fatigue were additionally reported as some of the most common side effects. In these studies, the maximum tolerable dose and plasma half-life were found to be of similar value. It was found that the dose-limiting toxicities were reversible and short-lived. No significant systemic toxicity was found for the biomarkers of organic functions. Under this protocol, tolerable and safe administered doses of NaSO_3_ were able to be administered. Gynecological patients comprised another phase 1 trial in which sodium selenite was also administered [121].

In other studies, from doses ranging from 500 to 50 µg, patients were administered NaSeO_3_. In these cases, the patient’s tolerable dose (maximum) was not reached. Carboplatin pharmacokinetics were not altered, and the drugs were well tolerated and safe when the carboplatin/paclitaxel and Se regimen were combined. As part of ongoing studies, a phase II clinical trial is suggested to use at 5000 µg dose. The Sangkyungwon Integrated Medical Center, radiation therapy and chemotherapy, combined with sodium selenite loadings of more than 5000 µg, revealed that terminal cancer patients could tolerate this level of dosing. Nontoxic levels were found at a 24 h time point after injection, namely, less than 260 µg ml^−1^ after this regimen, which involved 5000 µg and the additional involvement of radiotherapy and chemotherapy. A few cases of gastroenteric symptoms were discovered, but no adverse and irreversible events were detected in most cancer patients during this high loading of selenite or after the loading! Compared with that for normal cells, the accumulation of Se into cancer cells may be more toxic to these aberrant cells. The short half-life of the NaSeO_3_ may benefit chemotherapy, and Se may preferentially accumulate into cancer cells. In our case, drug resistance was experienced by patients under care as they were treated with chemotherapy in multiple forms. The quality of life was evaluated, as well as other parameters such as duration and dosage after a detailed segment modification in individual cases analyzed for durations of 3, 6, and 12 months were conducted [122]. Imaging was undertaken in some cases (Case 1): after a high dose of NaSO_3_, which was part of a multidisciplinary treatment, the pelvic region was imaged with CT (Figure 9). Here, there was evidence of advanced metastatic cancer in patients, a regimen of large doses of sodium selenite. After 18 months of chemotherapy and high doses of sodium selenite administered, there were dramatic optimistic changes. Where there once existed massive ascites (A), later, no sign of the disease was detected (B) for this particular case of peritoneal pelvic carcinomatosis (Figure 9).

As shown in Figure 10 below, the mechanism of action involves cell penetration for selenite when present; in the extracellular compartment, oxidation by selenite allows H_2_Se to be chemically converted and utilized. Through the multidrug transport system, the extracellular compartment can host cysteine; Cys was reduced through the intracellular action of H_2_Se on the cystine unit when inside the Xc-cystine/glutamate antiporter [122]. As the most reactive Se-containing compound commonly found in nature, H_2_Se can be absorbed in cancer cells in more significant amounts. Considering the GSH oxidation-reduction capacity, H_2_Se can be taken up more by drug-resistant tumor cells or metastatic tumors, leading to more significant toxicity imparted to the tumor. In cancer cells that experience more oxidative stresses, GSH is produced; GSH is a tripeptide, the middle residue of which is cysteine, and GSH is a stable form of soluble cysteine and intimately related to redox chemistry where we often consider the ratio between reduced (GSH) and oxidized glutathione (GSSG) as a biomarker. For advanced cancer cases, possible therapy can involve treatment with Se-containing compounds at high doses in some cases. In terms of cancer prevention and addressing the challenge of “chemoresistance” status in cancer patients who are in the advanced stages of the disease, and in combination with (i) radical surgery, (ii) radiotherapy, and (iii) chemotherapy, selenium may be a good choice because of the evidence that no severe adverse and lasting effects are found. Patients will eventually show drug resistance as they are treated with chemotherapy in a repetitive fashion. Patients showed drug resistance after being treated with several drugs. Terminal patients were among the subjects who were found to be chemo-resistant and to whom the sodium selenite was administered in high doses. A prominent combination of methods would be (i) radiation therapy with (ii) chemotherapy along with high doses of sodium selenite as a conditional and adaptable treatment. Immunotherapy, chemotherapy, irradiation, and surgery would encompass the standard treatment that could then involve modifications that could be simply tuned according to the duration of treatment and the dosage as well as the form of selenium (Figure 10). 

At institutions internationally, it will be necessary to test high doses of inorganic selenium in its forms through prospective clinical trials in the future.

## 8. Conclusions and Outlook

In conclusion, we have surveyed the literature, which also includes our original research work and reviewing work in the area. We began and will conclude this review optimistically as we look out to biochemical and also medicinal applications for the future. Selenium concentration, which bears the famous so-called “U-shaped toxicity” trend, might lead us to think that under no circumstances should we exceed upper limits. However, there are cases, especially in certain cancer trials, where when selenium was administered in extra-toxic doses, it did not appear to lead to any adverse effects. In this report, we reviewed the clinical and biomedical research. We sought to address what we propose as the selenium impacting the signaling pathways. So, our coverage of biochemistry, biology, and clinical undertakings should help other researchers better understand how to consider organic (e.g., methyl selenol) or inorganic selenium (e.g., sodium selenite) for future investigations, whether in the scientific laboratory setting or the chemotherapeutic treatment setting. We also had a chance to cover ferroptosis and senescence in addition to other topics.

As alluded to by what is included or excluded in Figure 6 and throughout the manuscript, the modification of signaling pathways by selenium-containing compounds is not yet overtly stated or clearly explained. Pertinent to the developing picture of SARS-CoV-2 and its effects on NRF2: selenium is not able to be indicated in this figure. How selenium-containing compounds modify KEAP signaling is a one-step jump in this reasoning. No research reports have emerged to our knowledge on selenium’s influence on KEAP directly. Indirectly, we propose NF-κB is related to KEAP. This current hypothesis needs further verification. However, as of yet, there is no research work on this specific topic. Although it is useful to suggest potential molecular mechanisms by which selenium, or selenium-containing molecules, can influence cell signaling, these mechanisms should be based on reported experimental data. For example, there is no evidence that selenium can influence KEAP function.

Finally, we want to provide ideas about future directions in this general field of research. We have provided a list of various suggestions for the community, which are briefly stated and grouped according to topic. The good news is that selenium biological and medical research, arguably being a unique outcropping of sulfur biological and medical research, continues to be developed; new compounds are constantly being investigated. This future outlook section is a combination of both our hopes and expectations. Every further finding is essential and a boon to the trace element research community.
Medicinal and Small molecule
There are small molecule chemistry studies that can be tested as cocktails.How chemical cocktails of synthetic Se-containing compounds can be prepared (chemical synthesis) and formulations and their synthesis, which contain selenium, are of research interest.A host of combinations of both reduced and oxidized forms of selenium can be tested in combination in cocktail form.New small synthetic molecules of selenium can be used in biological testing as fluorescent probes in biological studies. The molecules could be natural products in which the selenium is taking the place of the O or S atom, which is an isosteric position.How other trace elements or main group elements can be involved in facilitating the treatment of cancer and aging—elements such as fluorine, boron, gallium, etc.—is of interest. Compounds that are fluorescent, phosphorescent, or luminescent can be tested for their efficacy and for their ability to serve as chemosensors with proper selectivity and sensitivity; moreover, their biological activity can be determined.A deeper understanding of tellurium-containing compounds in which the tellurium center is isosteric with that of selenium and sulfur is of importance because the tellurium stays away from sulfur metabolism because Te is comparatively more metallic. The understanding of tellurium compounds will impact further understanding of selenium-containing compounds and their capacity. As a warning, the Te–C bonds are weaker, and, therefore, the research is more challenging.Biophysical and Pharmacological
The field shall require more biophysical chemistry studies, protein-related studies in the form of mutation studies with selenium-containing proteins, and novel unnatural selenium-containing AAs.As mentioned above, the assessment of the pharmacokinetics of new compounds will be essential and helps quantify the performance of the compounds.Biology and Biochemistry
Further exploration of the (side) effects of Se and Se-containing small molecules on biology is required. One goal of these medicinal studies would be to check the relationship between the induction of Akt activity and the reduction of the ASK1 complex.The pro/antioxidant effect of selenite molecular action can be better defined and more clearly researched under a variety of biological conditions.How metabolic engineering can be used to help prepare further synthetic compounds of selenium is an important branch of this study.Clinical and Animal Testing
It will be incredibly important to carry out further in vivo studies of MSeC.MSeC studies that can help assess the maximum tolerable dose and the half-life of the concentration of MSeC in human biology will be greatly welcomed by the community.Because MSeC is commercially available as a readily available over-the-counter supplement form, the application of efficacy of treating cancer in humans with this chemical would be an interesting investigation.As mentioned above, the NRF2 activators and Keap1 modification incurred by NF-κB activation would be a great focus of study. An exact understanding of selenium involvement in these processes would be important to investigate to ascertain whether selenium-containing compounds may be involved in altering NRF2 activity; this could be achieved by reducing NF-κB or modifying Keap1.GPX3 from the blood of cancer patients should be monitored, as well as the influence of selenium supplementation can be better defined in clinical studies.During the hospital stays (intensive care unit stay duration), the duration of *ventilation* by mechanical means and the overall survival rate were not altered by selenium intervention so far. Therefore, the possible positive clinical effectiveness can be observed upon completing, summarizing, and analyzing more clinical trials when they become available [105].High research standards will have to be involved in intervention studies that better scrutinize a proposed underlying feed-forward mechanism because of our observed association of Se deficiency with mortality risk. Safety and limited expenses are suitable for adjuvant treatments.Se supplementation with patients enduring ARDS can be sought, as well as other clinical applications which might be applicable and accessible. Hypothetical mechanisms can be proposed for Se supplements that would then prevent COVID-19 [123].Environmental
The effect on human health (possibly positive) of the prevalence of selenium-containing material in common environmental and commercial forms such as CdSe can be explored. In particular, there are different forms (crystal forms, bulk, and nanocrystalline) of, e.g., CdSe, that could be studied for their biological activity. Their ability and mechanism to enter the biological system and the rate at which it can do it and also later be eliminated would be important biological data for the community.

## Figures and Tables

**Figure 1 molecules-28-05234-f001:**
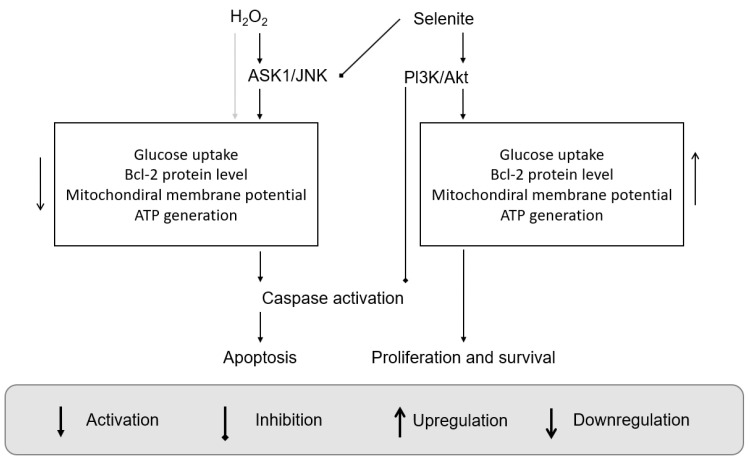
A schematic diagram of the effects of H_2_O_2_ and selenite on cell death and survival. Physiological level of selenite increases cell proliferation and survival by blocking apoptosis induced by H_2_O_2_ through inhibition of ASK1/JNK and activation of PI3K/Akt pathways (Reprinted/adapted with permission from reference [6] 2002, Yoon, S.-O).

**Figure 2 molecules-28-05234-f002:**
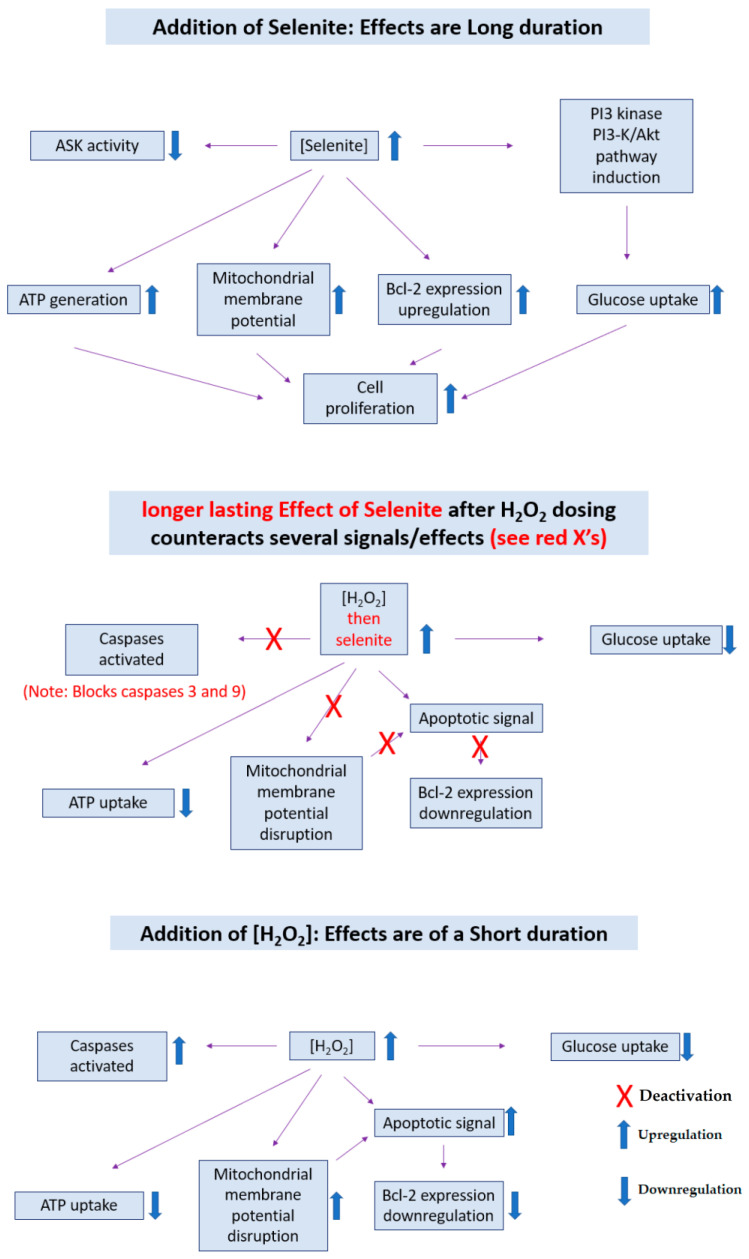
The roles of (top, middle) selenite (longer term) and (middle, bottom) H_2_O_2_ addition (shorter term) on the various aspects of the biological pathways (adapted from reference [6]).

**Figure 3 molecules-28-05234-f003:**
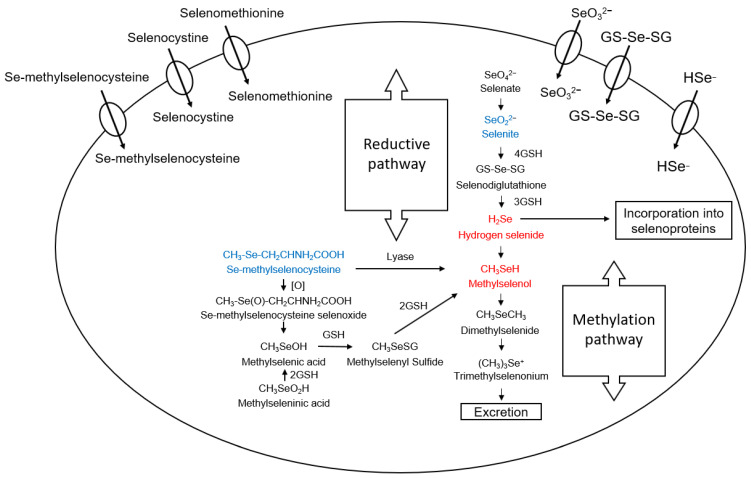
A description of selenium metabolism and activation, including absorption and excretion. Specifically, the extracellular concentration of the selenium compounds can enter the cell and participate in reactions. We understand this as a range of biochemistry that ultimately leads to the production of selenium proteins. The reduction and methylation pathways are distinguished, and the presence of GSH is shown. All names of the selenium compounds are provided, too (adapted from reference [2]. See reference [8] for a fuller account and additional information]) Note: when conceptualizing the action of GPX and TrxR as selenium enzymes (see below), they should be separated and distinguished from those of glutathione and thioredoxin (as small redox proteins) in the regulation of selenium metabolism in the cell.

**Figure 4 molecules-28-05234-f004:**
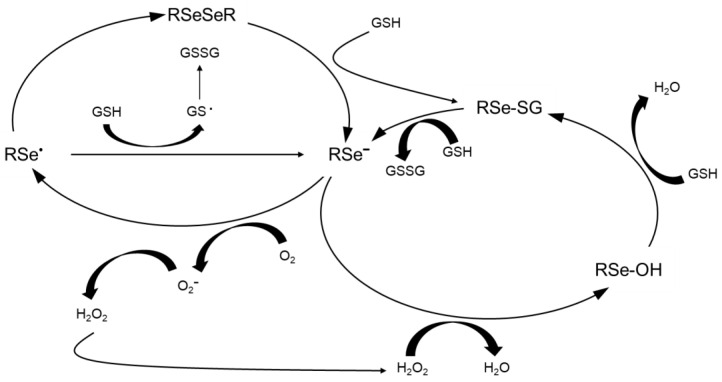
Production of ROS by selenium (adapted from reference [2]).

**Figure 5 molecules-28-05234-f005:**
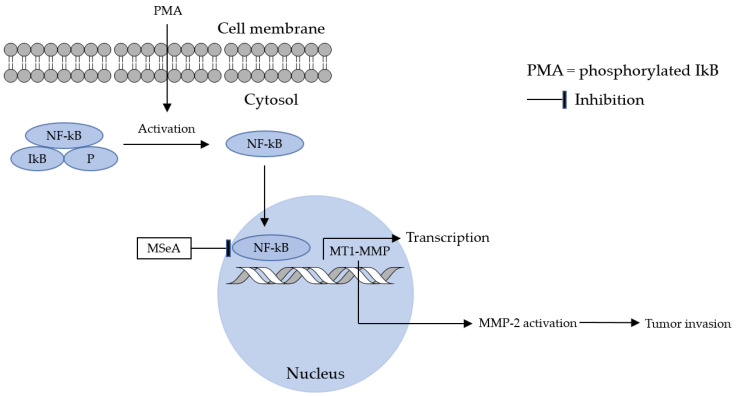
NF-kB binds to the IkB active form (phosphorylation). MSeA blocks the activation of NF-kB and the translocation of NF-kB. In other words, MSeA inhibits the IκBα and p65 phosphorylation by PMA, thereby blocking the nuclear translocation of NF-κB.

**Figure 6 molecules-28-05234-f006:**
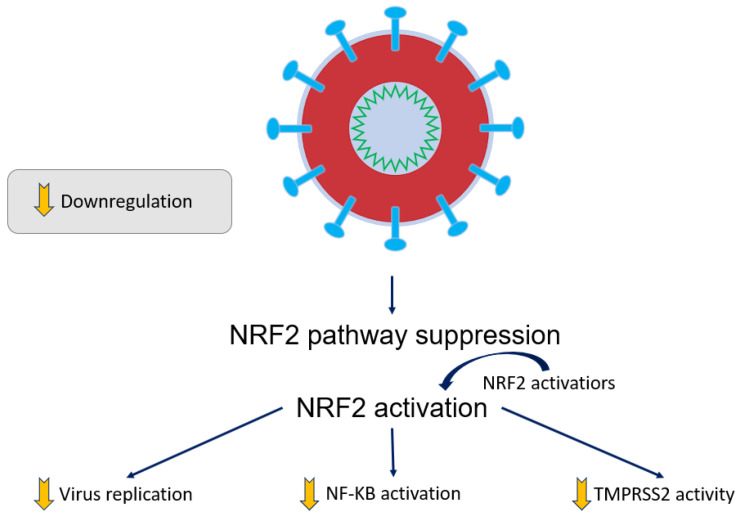
Simple depiction of the SARS-CoV-2 virus and its role in NRF2 activation and replication, as well as NF-κB activation and TMPRSS2 activity (adapted from reference [55]).

**Figure 7 molecules-28-05234-f007:**
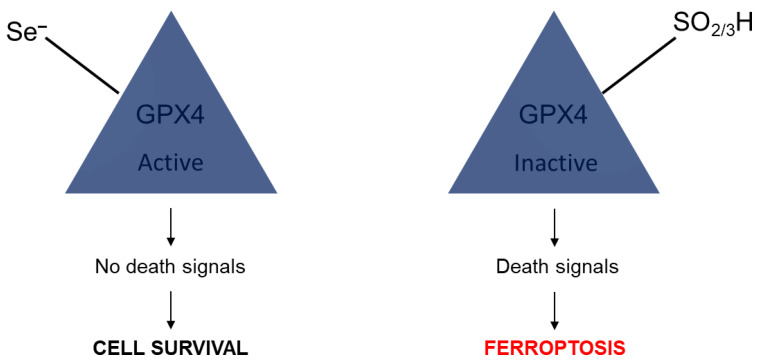
Pathways of ferroptosis and cell survival. Adapted from the graphical abstract of reference [97].

**Figure 8 molecules-28-05234-f008:**
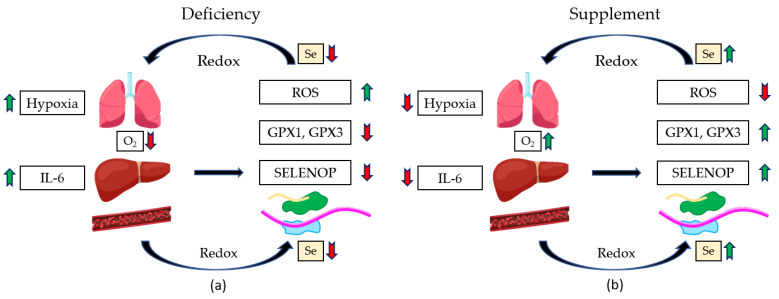
In the case of ARDS, Se treatment increased Se level, SELENOP, GPX1, GPX3, antioxidant capacity, and O_2_ flow but decreased ROS and IL-6. A two-part cartoon (reprinted/adapted from reference [105] 2020, Moghaddam, A.) in which the left side (**a**) is showing a deficiency of selenium. The right side (**b**) shows the effects of selenium addition (supplementation). In each half, the redox cycle is shown for human organs (left side in each half) and biochemistry (right side of each half).

**Figure 9 molecules-28-05234-f009:**
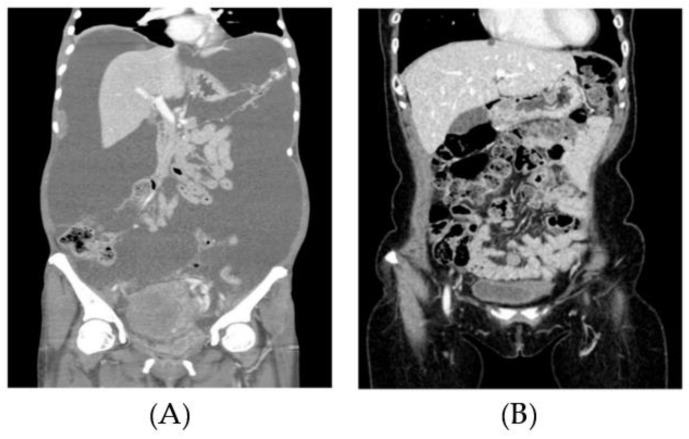
CT image of the pelvic region of a patient. Peritoneal carcinomatosis with massive ascites (**left**, (**A**)) and no evidence of disease after 18 months of chemotherapy (**right**, (**B**)), interval debulking surgery, and PARP (Poly (ADP-ribose) polymerase) inhibitor maintenance therapy with high dose sodium selenite treatment. “Reprinted/adapted directly from reference [2]. 2021, Kim, S.J”.

**Figure 10 molecules-28-05234-f010:**
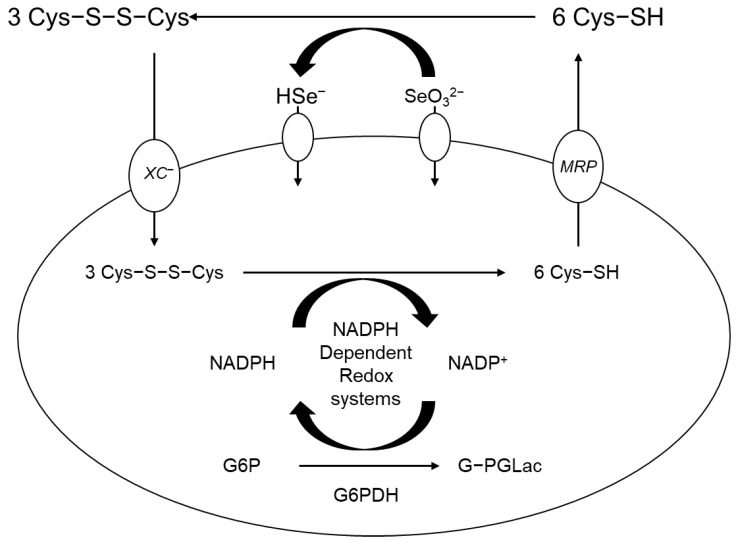
Absorption mechanism of high dose of sodium selenite (adapted from reference [2]).

**Table 1 molecules-28-05234-t001:** The normal range of blood values and tolerable intake values, and daily intake values over ages from internet sources.

Criteria	Values	Notes	References
Normal range of blood selenium	120–160 μg/L	“The Tolerable Upper Intake Level (UL) for selenium for all adults 19+ years of age and pregnant and lactating women is 400 micrograms daily; a UL is the maximum daily intake unlikely to cause harmful effects on health.”	[3]
Tolerable intake, upper level(s)	400 μg per day		[3]
Daily required intake for human	See notes	0–3 years of age: 10–20 micrograms (mcg) per day. 4–6 years of age: 20 mcg per day. 7–10 years of age: 30 mcg per day. Adolescent or adult males: 40–70 mcg per day. Adolescent or adult females: 45–55 mcg per day.	[4]

## Data Availability

Not applicable.

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
