# Peer review of "Nontoxic Levels of Se-Containing Compounds Increase Survival by Blocking Oxidative and Inflammatory Stresses via Signal Pathways Whereas High Levels of Se Induce Apoptosis"

_molecules, 2023, doi:10.3390/molecules28135234_

Round 1

Reviewer 1 Report

In the manuscript, the authors describe the antioxidant and anti-inflammatory action of selenite in the entire spectrum of pathological conditions. The part related to Covid-19 is particularly interesting. Some interventions are needed to make the presented facts easier to follow (especially part 2).

In the Introduction, from the second paragraph, the antiapoptotic effect of selenium is explained. In Figure 1, the last row should be presented more clearly and larger. Mention which conditions regarding the level and chemical form of selenium...etc determine the antiapoptotic effect of selenium. It is known that selenium can have a proapoptotic effect in numerous cases, so this should be mention/discussed as well (briefly).

On pg 4 methyl selenocysteine should be considered/named as an amino acid (an abbreviation should also be introduced there). It should also be checked whether all other abbreviations are introduced at the first mention. This should be checked through the entire manuscript.

Is it antitumor capacity of listed forms of selenium linked with selenium proapoptotic effects; more clearly emphasized and explained (ln 93-98)?

Glutathione peroxidase (GPx) and thioredoxin reductase (TrxR) as selenium enzymes should be separated/distinguish from glutathione and thioredoxin (as small redox proteins) in the regulation of selenium metabolism in the cell (ln 103, 104, 105). For easier understanding of the text and a better flow, the legend for Figure 3 should be more detailed/more informative. I would recommend a very useful paper: Burk RF, Hill KE. Regulation of Selenium Metabolism and Transport. Annu Rev Nutr. 2015;35:109-34

In part 3 (ln 178) should it be EXM instead of ECM?

Link between caspase activation and integrins (events on the cell surface and in the cell)? Reference 48 should be adequately cited.

Pg 6, from ln 179 reference should be cited ref 40 as describes mode of methylseleninic acid action and cite your references whose results you refer to (ln 185-186)!   

Ln 188 Why is [MSeA] in parentheses?

On pg 8 ln 245 and 250 references should be provided.

In several places, the text is confusing, and I had difficulty following the abbreviations!

In the Part 4  the anti-inflammatory effects of selenium and the points of its action are more systematically presented the text flow is more clear. ln 333 remove the full stop.

For Figur 7: the influence of selenium supplementation is well described in the text, it is not clear to me what the redox arrows represent (is it redox status in the cell or blood?), if it can be clarified.

In part 6 the activity of GPx3 from the blood of cancer patients should be mentioned, as well as the influence of selenium supplementation!

In the part 7 the highlight from the first part of the manuscript, which refers to the pro/antioxidant effect of selenite molecular action is missing (like bullet 5).

Author Response

-Comment: “Some interventions are needed to make the presented facts easier to follow (especially part 2).”

Response:  We have made a great number of changes to the text to smooth it over and to help present facts more easily. We have also added more vitally important references and figures that will help with the readability of the material and to help the readership more easily find and understand the information.

-Comment: In Figure 1, the last row should be presented more clearly and larger. Mention which conditions regarding the level and chemical form of selenium...etc determine the antiapoptotic effect of selenium. It is known that selenium can have a proapoptotic effect in numerous cases, so this should be mention/discussed as well (briefly).

Response: Figure 1 has now been modified more clearly. The last “row” which seemed like it was part of the diagram was in fact a key, a guide to the arrows used.

-Comment: On pg 4 methyl selenocysteine should be considered/named as an amino acid (an abbreviation should also be introduced there). It should also be checked whether all other abbreviations are introduced at the first mention. This should be checked through the entire manuscript.

Response: The abbreviation MSeC for methyl selenocysteine was added on page 4, In 88.

-Comment: Is it antitumor capacity of listed forms of selenium linked with selenium proapoptotic effects; more clearly emphasized and explained (ln 93-98)?

Response: The anti-tumor capability is described in Figure 1.

-Comment: Glutathione peroxidase (GPX) and thioredoxin reductase (TrxR) as selenium enzymes should be separated/distinguish from glutathione and thioredoxin (as small redox proteins) in the regulation of selenium metabolism in the cell (ln 103, 104, 105). For easier understanding of the text and a better flow, the legend for Figure 3 should be more detailed/more informative. I would recommend a very useful paper: Burk RF, Hill KE. Regulation of Selenium Metabolism and Transport. Annu Rev Nutr. 2015;35:109-34

Response:  We have cited this excellent article. We have expanded the caption to help describe the goings on in the graphic. We have also added text to better elucidate their message.

-Comment: In part 3 (ln 178) should it be EXM instead of ECM?

Response: ECM means extracellular matrix. Therefore, the ECM is appropriate and we have also placed the full name there.

-Comment: Link between caspase activation and integrins (events on the cell surface and in the cell)? Reference 48 should be adequately cited.

Response: For proper use of the text's content and references, the position of reference 48 was moved to ln 178.

-Comment: Pg 6, from ln 179 reference should be cited ref 40 as describes mode of methylseleninic acid action and cite your references whose results you refer to (ln 185-186)!   

Response: Reference 40 was added to pages 6 (In 179).

-Comment: Ln 188 Why is [MSeA] in parentheses?

Response:  The brackets are a rotation which define the concentration of a substance in solution.

-Comment: On pg 8 ln 245 and 250 references should be provided.

Response: Reference 49 is related and was added to ln 245 and 250.

-Comment: In several places, the text is confusing, and I had difficulty following the abbreviations!

In the Part 4 the anti-inflammatory effects of selenium and the points of its action are more systematically presented the text flow is more clear. ln 333 remove the full stop.

Response: The full stop of line 333 was modified to a comma.

-Comment: For Figure 8: the influence of selenium supplementation is well described in the text, it is not clear to me what the redox arrows represent (is it redox status in the cell or blood?), if it can be clarified.

Response: We fully revised the figure so that it is presented more clearly.

-Comment: In part 7 the activity of GPX3 from the blood of cancer patients should be mentioned, as well as the influence of selenium supplementation!

Response: We have added this information into the text and have also added a future comment about this.

-Comment: In the part 8 the highlight from the first part of the manuscript, which refers to the pro/antioxidant effect of selenite molecular action is missing (like bullet 5).

Response: We have added this information into the text and have also expanded on the future comment about this near the end of the article.

Reviewer 2 Report

Manuscript ID molecules-2367461 by Jongkeol An , An-Sik Chung and David G. Churchill , entitled "Nontoxic levels of Se compounds increase survival by blocking oxidative and inflammatory stresses via signal pathways whereas high levels of Se induce apoptosis", reviews the biology of selenium-containing compounds in the contexts of cancer and viral infections. Although this subject matter is likely to have broad appeal, the manuscript is written in a confusing, anecdotal and vague style, and contains multiple technical and grammatical errors. Consequently, the following points should be addressed before this narrative review can be considered suitable for publication in MDPI Molecules:

MAJOR POINTS:

1) The standard of scientific writing should be improved, as described in the following section.

2) Please adhere to the standard nomenclature for the naming of selenoproteins, as approved by the HUGO Gene Nomenclature Committee (PMID: 27645994).

3) Throughout the manuscript, the modification of signalling pathways by selenium-containing compounds is not overtly stated or clearly explained. An example of this is Figure 6, on SARS-CoV2 and its effects on Nrf2: selenium is not indicated anywhere in this figure. After reading this review twice, I am still not sure about how selenium-containing compounds modify KEAP signalling, etc.

4) None of the figure legends contain enough information for their associated figures to be clearly understood. For example, the legend of Figure 8 does not indicate what the two separate panels ((A) and (B)) show.

5) Section 7, Outlook and Conclusions, is very vague and anecdotal. It does not give any specific ideas about future directions in this field of research.

MINOR POINTS:

1) Lines 87-88: selenium methionine (do the authors mean selenomethionine?) is not a coded amino acid. It is though to be formed by reaction of methionine with selenium-containing compounds.

2) Define non-standard abbreviations on first use. For example, line 90, what does "MSec" stand for?

3) Lines 174-175: fibronectin and vitronectin are not integrins. They are integrin ligands.

4) The normal range of blood selenium is not mentioned until Section 5. The Tolerable Upper Intake Levels are not stated. The effects of excess selenium intake are not discussed. Surely these are important aspects for the Introduction?

5) There are several minor typographical errors.

Although the standard of English in this review is reasonably good, the quality of scientific writing is poor. Many aspects are difficult to understand because of this, reducing the potential impact of this work. Specific points are:

1) Many of the sentences do not make any sense, eg. the sentences beginning "The first receptor-mediated oxidation can be measured as a half-life, ..." (lines 342-344); and "Safety and limited expenses are suitable for adjuvant therapy." (lines 406-407).

2) Other sentences are incomplete or refer to subsequent information that is not given. For example, for the sentence "As summarized in the literature, redox signaling in immune cells and calcium and Se level dependencies are described below." (lines 344-346). This is another poorly constructed sentence, which would leave the reader expecting citations of summaries in the literature (not given) and further discussion of calcium (this is the last mention of either "calcium" or "Ca2+").

3) Much of the language is vague or anecdotal. As a result, it may be difficult for a non-native English speaker to understand some of the concepts. For example, in the sentence "Understanding of tellurium (the downstairs neighbor of Se) research..." (line 567), I would guess the "downstairs" part is referring to the relative positions in the Periodic Table...but this might not be obvious to every reader.

Author Response

-Comment: 1) Although this subject matter is likely to have broad appeal, the manuscript is written in a confusing, anecdotal and vague style, and contains multiple technical and grammatical errors. Consequently, the following points should be addressed before this narrative review can be considered suitable for publication in MDPI Molecules:The standard of scientific writing should be improved, as described in the following section.

Response: We have revised the entire manuscript to bring the standard of scientific writing up to a higher level. If you should need more details about the minor changes that we made, please let us know and we can provide earlier drafts with tracked changes.

-Comment: 2) Please adhere to the standard nomenclature for the naming of selenoproteins, as approved by the HUGO Gene Nomenclature Committee (PMID: 27645994).

Response: In consideration of the HUGO Gene Nomenclature Committee, 'GPx' was modified to 'GPX'.

-Comment: 3) Throughout the manuscript, the modification of signalling pathways by selenium-containing compounds is not overtly stated or clearly explained. An example of this is Figure 6, on SARS-CoV2 and its effects on Nrf2: selenium is not indicated anywhere in this figure. After reading this review twice, I am still not sure about how selenium-containing compounds modify KEAP signalling, etc.

Response: We have tried to make this clear by changing the caption adding additional text into the manuscript.

-Comment: 4) None of the figure legends contain enough information for their associated figures to be clearly understood. For example, the legend of Figure 9 does not indicate what the two separate panels ((A) and (B)) show.

Response: In Figure 9, we added a description of the differences in two separate panels (A), (B).

-Comment: 5) Section 8, Conclusion and Outlook, is very vague and anecdotal. It does not give any specific ideas about future directions in this field of research.

Response: We have installed a more proper conclusion.  Also, we have reedited the future outlook and made it more specific, in some cases, and better stated to help research.

MINOR POINTS:

-Comment: 1) Lines 87-88: selenium methionine (do the authors mean selenomethionine?) is not a coded amino acid. It is though to be formed by reaction of methionine with selenium-containing compounds.

Response: selenium methionine was modified to selenomethionine.

-Comment: 2) Define non-standard abbreviations on first use. For example, line 90, what does "MSec" stand for?

Response: MseC stands for methyl selenocysteine. The abbreviation MSeC for methyl selenocysteine was added on page 4, In 88. We used the full terminology and placed the abbreviation in parentheses. Also, we prepared a full abbreviation list of the end of the paper.

-Comment: 3) Lines 174-175: fibronectin and vitronectin are not integrins. They are integrin ligands.

Response: In lines 174-175, intergrin was modified to Integrin ligand. We changed the wording in this text to make the meaning more sensible.

-Comment: 4) The normal range of blood selenium is not mentioned until Section 6. The Tolerable Upper Intake Levels are not stated. The effects of excess selenium intake are not discussed. Surely these are important aspects for the Introduction?

Response: We have now addressed these concerns. We have placed more information in the introduction. In addition, we have provided a short Table to make these important values stand out more.

-Comment: 5) There are several minor typographical errors.

Response:  We have revised the review from beginning to the end. 

-Comment: Although the standard of English in this review is reasonably good, the quality of scientific writing is poor. Many aspects are difficult to understand because of this, reducing the potential impact of this work. Specific points are:

Response:  We have made many changes throughout the manuscript to change these issues

-Comment: 1) Many of the sentences do not make any sense, eg. the sentences beginning "The first receptor-mediated oxidation can be measured as a half-life, ..." (lines 342-344); and "Safety and limited expenses are suitable for adjuvant therapy." (lines 406-407).

Response:  we have changed these specific issues and have also gone on to revise the entire work from the beginning to the end.

-Comment: 2) Other sentences are incomplete or refer to subsequent information that is not given. For example, for the sentence "As summarized in the literature, redox signaling in immune cells and calcium and Se level dependencies are described below." (lines 344-346). This is another poorly constructed sentence, which would leave the reader expecting citations of summaries in the literature (not given) and further discussion of calcium (this is the last mention of either "calcium" or "Ca2+").

Response: We have attended to this remark.

-Comment: 3) Much of the language is vague or anecdotal. As a result, it may be difficult for a non-native English speaker to understand some of the concepts. For example, in the sentence "Understanding of tellurium (the downstairs neighbor of Se) research..." (line 567), I would guess the "downstairs" part is referring to the relative positions in the Periodic Table...but this might not be obvious to every reader.

Response:  As mentioned above, we have taken time to change the text; many changes that were made throughout the manuscript how help provided a smoother submission and we feel that the readability has greatly improved now.

Reviewer 3 Report

An et al. submitted a review article on selenium compounds and their metabolism, signaling pathway, and health-related issues. This is an interesting review. Addressing the following general points shall strengthen this review manuscript.

The idea of selenium effect focusing solely or mainly on apoptosis is outdated and fragmented. To control cell growth/eliminated unfavorable cells (e.g., cancer/precancerous cells), results in the last decade show at least two new pathways: senescence and ferroptosis. For senescence, examples of papers of such include PMID: 20157118; 20709753. ATM pathway, senescence/nano-selenium species, and redox regulation can be placed into the context and added in the existing figures and in the relevant text. The same is true for ferroptosis (PMID: 34931062; 29854274; 29290465).

Figure 3 implies that the organic and inorganic selenium species enter cells through transporter proteins. This is largely assumed as how selenium compounds cross the membrane is largely unknown. Revision should be appropriately made.

Author Response

-Comment: The idea of selenium effect focusing solely or mainly on apoptosis is outdated and fragmented. To control cell growth/eliminated unfavorable cells (e.g., cancer/precancerous cells), results in the last decade show at least two new pathways: senescence and ferroptosis. For senescence, examples of papers of such include PMID: 20157118; 20709753. ATM pathway, senescence/nano-selenium species, and redox regulation can be placed into the context and added in the existing figures and in the relevant text.

Response: We have added text to this effect. Please see the new section in our manuscript that describes the 5 papers listed above. We also provide a figure to help show the relationship to the oxidation of lipids that ensue from the ferroptosis.

-The same is true for ferroptosis (PMID: 34931062; 29854274; 29290465).

Response: Ferroptosis is now described in the text according to these three references. The three references have been added.

Response: -Figure 3 implies that the organic and inorganic selenium species enter cells through transporter proteins. This is largely assumed as how selenium compounds cross the membrane is largely unknown. Revision should be appropriately made.

Response: Selenomethionine and other selenium amino acids can be transported into cells as amino acids transport systems and selenite reacts with sulfhydryl groups, which can be transported into cells. Selenite oxidizes sulfhydryl groups such as glutathione (GSH) and selenodiglutathione can be transported into cells. These are the general mechanisms about selenium compounds and how they can be transported as known by biologists.

Round 2

Reviewer 2 Report

The revised manuscript still contains numerous errors, some of which are major and likely to confuse or mislead readers:

1) In lines 109-110, it is stated that “selenomethionine (SeM); SeM is considered the 21st encoded amino acid.”. This is incorrect. Selenocysteine, not SeM, is considered to be the 21st encoded amino acid. SeM can be misincorporated into proteins in place of methionine, or can be formed post-translationally. Also, in line 804, the abbreviation SeM is defined incorrectly as selenocysteine.

2) Although it is useful to suggest potential molecular mechanisms by which selenium, or selenium-containing molecules, can influence cell signalling, these mechanisms should be based on reported experimental data. For example, there is no evidence that selenium can influence KEAP function. Such conjecture should be restricted to the Conclusions and Outlook section, or avoided altogether.

3) Figure 2: is not fully explained by its associated legend, eg. what do the different arrows and crosses represent? Figure 4: lacks a figure legend. Figure 5: lacks a figure legend; “Transcirption” is not spelled correctly. Figure 6: lacks a figure legend. Figure 7: lacks a title and a complete figure legend. Figure 8: the reference is incorrectly cited, i.e. should “(Nutrients 2020)” be “[103]”? Figure 9: the title is misleading; there are “massive ascites” shown in panel (A), rather than “no sign of the disease detected”.

4) The use of abbreviations is still inconsistent. Once an abbreviation has been defined, use it throughout the rest of the manuscript.

5) Line 652. GSH is not a form of cysteine, even if it is derived from this amino acid.

6) The Conclusions and Outlook section contains many incomplete ideas. For example, “Compounds that are fluorescent, phosphoresce or luminescent can be tested”. (“phosphoresce” should be “phosphorescent”). Tested for what? I understand that such compounds might be useful as markers for tumours (presuming that they accumulate in cancer cells), but this is not explained. There is a lack of explanation of most other points in this section.

7) The input of reviewers should be indicated in the Acknowledgements section, rather than in the main body of text.

8) Some of the reference citations are incorrect, eg. line 747, the reference cited ([107]) has no relation to Covid-19 and selenium.

The writing style remains very confused and disorganized, making it very difficult for readers to understand. I recommend a complete rewrite by a native English speaker.

There is a lack of logical flow to the progression of ideas. Many of the sentences are incomplete, leaving the reader wondering what the next point is, eg. the paragraph ending on line 612 ends with “Gynecological patients comprised another phase 1 trial in which Na selenite was administered”, and there is no further discussion of the outcomes or their relationship to the rest of this review; line 295-296, “Oxidizing the sulfhydryl group.” is not a complete sentence; “The NOX2 case is a clear case” is not an informative sentence.

The language used is anecdotal in many places. For example, line 729, the phrase “downstairs neighbor of Se” highlighted in my original review, was not replaced.

There are numerous minor typographical errors throughout.

Author Response

Comment: In lines 109-110, it is stated that “selenomethionine (SeM); SeM is considered the 21st encoded amino acid.”. This is incorrect. Selenocysteine, not SeM, is considered to be the 21st encoded amino acid. SeM can be misincorporated into proteins in place of methionine, or can be formed post-translationally. Also, in line 804, the abbreviation SeM is defined incorrectly as selenocysteine.

Response: In lines 116-119, we corrected “SeC is considered the 21st encoded amino acid”. We corrected the abbreviation SeM (selenomethionine) in line 864.

Comment: Although it is useful to suggest potential molecular mechanisms by which selenium, or selenium-containing molecules, can influence cell signaling, these mechanisms should be based on reported experimental data. For example, there is no evidence that selenium can influence KEAP function. Such conjecture should be restricted to the Conclusions and Outlook section, or avoided altogether.

Response: We moved the speculative paragraph to the conclusions section.

Comment: Figure 2: is not fully explained by its associated legend, eg. what do the different arrows and crosses represent? Figure 4: lacks a figure legend. Figure 5: lacks a figure legend; “Transcirption” is not spelled correctly. Figure 6: lacks a figure legend. Figure 7: lacks a title and a complete figure legend. Figure 8: the reference is incorrectly cited, i.e. should “(Nutrients 2020)” be “[103]”? Figure 9: the title is misleading; there are “massive ascites” shown in panel (A), rather than “no sign of the disease detected”.

Response: We added a figure legend to Figure 2. We think it is clearly explained without figure legends in Figure 4. We corrected the typo in Figure 5 and added a figure legend. We added a figure legend to Figure 6. In Figure 7, the title was modified and the Figure was abbreviated therefore no legend was supplied. In Figure 8, we corrected the reference. We corrected the title of Figure 9.

Comment: The use of abbreviations is still inconsistent. Once an abbreviation has been defined, use it throughout the rest of the manuscript.

Response: We revised the manuscript so that the use of abbreviations is more consistent, and only abbreviations are used after abbreviations are defined.

Comment: Line 652. GSH is not a form of cysteine, even if it is derived from this amino acid.

Response: We clarified the definition of GSH and we defined it clearly in the abbreviations section.

Comment: The Conclusions and Outlook section contains many incomplete ideas. For example, “Compounds that are fluorescent, phosphoresce or luminescent can be tested”. (“phosphoresce” should be “phosphorescent”). Tested for what? I understand that such compounds might be useful as markers for tumors (presuming that they accumulate in cancer cells), but this is not explained.

Response: We have clarified this sentence about fluorescent molecules. Moreover, we have restructured the entire section to give it a kind of superstructure, and it should appear more logical than before but still appear as a “to do” list. We have rewritten certain sentences in this section too to make them clearer.

Comment: There is a lack of explanation of most other points in this section.

Response: We have now restructured and rewritten this section.

Comment: The input of reviewers should be indicated in the Acknowledgements section, rather than in the main body of text.

Response: We moved the mentioning of the reviewers to the Acknowledgements section.

Comment: Some of the reference citations are incorrect, eg. line 747, the reference cited ([107]) has no relation to Covid-19 and selenium.

Response: We changed this reference and added two references here that are pertinent to COVID.

Comment: The writing style remains very confused and disorganized, making it very difficult for readers to understand. I recommend a complete rewrite by a native English speaker.

Response: We have taken to heart this comment and a native speaker has edited and also proofread the manuscript again. If the reviewer has specific comments, we would be happy to attend to specific remarks.

Comment: There is a lack of logical flow to the progression of ideas. Many of the sentences are incomplete, leaving the reader wondering what the next point is, eg. the paragraph ending on line 612 ends with “Gynecological patients comprised another phase 1 trial in which Na selenite was administered”, and there is no further discussion of the outcomes or their relationship to the rest of this review; line 295-296, “Oxidizing the sulfhydryl group.” is not a complete sentence; “The NOX2 case is a clear case” is not an informative sentence.

Response: We have fixed the short and incomplete sentences mentioned above. We have tried to create a better logical flow for example by adding topic sentences and reading out loud.

Comment: The language used is anecdotal in many places. For example, line 729, the phrase “downstairs neighbor of Se” highlighted in my original review, was not replaced.

Response: We have tried to find all anecdotal language and fix it. We changed the “downstairs” phrase.

Comment: There are numerous minor typographical errors throughout.

Response: We have proofread the article again and fond many small errors. The manuscript should be in excellent shape now concerning this comment.

Reviewer 3 Report

The authors have adequately addressed the comments raised in the previous round of review.  Some additional edits may be needed.

1. The reference list and citations may need to be taken another look. For example, reference 52 starts with "and" rather with a name.

2. It appears that Figure 7 is very similar to the original illustration being published in cell. In particular, the hanging moon shapes seem exactly the same as those being published and may evoke copyright issues. Redraw. 

Author Response

Comment: The reference list and citations may need to be taken another look. For example, reference 52 starts with "and" rather with a name.

Response: We checked once again the reference list including reference 51.

Comment: It appears that Figure 7 is very similar to the original illustration being published in cell. In particular, the hanging moon shapes seem exactly the same as those being published and may evoke copyright issues. Redraw. 

Response: We removed the hanging moon shapes and redrew and also abbreviated the figure to show only the lower half that underscores the effect of the mutation.